# CROWD-CALIBRATOR: Can Annotator Disagreement Inform Calibration in Subjective Tasks?

**Urja Khurana**[🌷], **Eric Nalisnick**[🦀], **Antske Fokkens**[🌷], **Swabha Swayamdipta**[🧽]

[🌷] Computational Linguistics and Text Mining Lab, Vrije Universiteit Amsterdam
[🦀] Department of Computer Science, Johns Hopkins University
[🧽] Thomas Lord Dept. of Computer Science, University of Southern California
{u.khurana,antske.fokkens}@vu.nl, nalisnick@jhu.edu, swabhas@usc.edu

## Abstract

Subjective tasks in NLP have been mostly relegated to objective standards, where the gold label is decided by taking the majority vote. This obfuscates annotator disagreement and the inherent uncertainty of the label. We argue that subjectivity should factor into model decisions and play a direct role via calibration under a selective prediction setting. Specifically, instead of calibrating confidence purely from the model's perspective, we calibrate models for subjective tasks based on crowd worker agreement. Our method, CROWD-CALIBRATOR, models the distance between the distribution of crowd worker labels and the model's own distribution over labels to inform whether the model should abstain from a decision. On two highly subjective tasks, hate speech detection and natural language inference, our experiments show CROWD-CALIBRATOR either outperforms or achieves competitive performance with existing selective prediction baselines. Our findings highlight the value of bringing human decision-making into model predictions.

## 1 Introduction

Natural language is inherently subjective, leading to subjectivity in classification tasks (Aroyo & Welty, 2015; Plank, 2022; Cabitza et al., 2023; Jamison & Gurevych, 2015; Pavlick & Kwiatkowski, 2019). Yet, in natural language processing, most such tasks are treated as if there exists *a single ground truth*. The conventional setup consists of a small number of annotators labeling each sample and taking the majority vote determines the final label. However, this setup dismisses subjectivity in implications (for e.g., in NLI; Pavlick & Kwiatkowski, 2019), and removes minority voices (for e.g., in safety-critical applications like hate speech detection; Prabhakaran et al., 2021; Sap et al., 2022). While there are many cases for which humans are more likely to agree with each other (Jiang et al., 2021a; Salminen et al., 2019), there are also cases where there is a lack of consensus (Khurana et al., 2022). In such cases, models, rather than predicting a single label, must make decisions that reflect potential *disagreements*. Selective prediction frameworks (Geifman & El-Yaniv, 2017; Chow, 1957) allow for this kind of model abstention. In this paper, we argue that selective prediction is an ideal fit for subjective tasks.

Ideally, we would want a model's confidence, i.e. softmax probabilities, for its prediction to reflect human disagreement. However, neural models with a large number of parameters tend to be overconfident (Guo et al., 2017; Chen et al., 2023). A straightforward way to make a model aware of human variation in the label is to use *soft labels* (Jamison & Gurevych, 2015; Uma et al., 2020; 2021b). Instead of a model predicting *one* label (hard label), we want the model to output the human label distribution. However, most available NLP datasets only provide 3-5 annotations per sample, which limits the distributions that the model can learn. We need many more annotations per sample to *learn* the crowd's beliefs.

How can we still have our model make human subjectivity-aware predictions? We propose CROWD-CALIBRATOR, a method that calibrates models for subjective tasks according to

crowd disagreement. Most calibrators either rely on the model's confidence or have a separate calibrator that outputs if a model is correct. For subjective tasks, there are limited ways of calibrating a model as mostly there is no one correct label. Our CROWD-CALIBRATOR determines if the model's output distribution is close to the human judgment distribution. If that is the case, then the model makes a prediction. If the model is far from the human distribution, then the model *abstains*.

We apply CROWD-CALIBRATOR to two subjective NLP tasks: hate speech detection and natural language inference (NLI), a task for which an abundance of human subjectivity data is available (Nie et al., 2020b). We show the potential of our setup by comparing it to selective prediction baselines such as MaxProb and Kamath et al. (2020). Our method is competitive with these baselines for hate speech detection and outperforms them for NLI. We also show that our setup beats baselines on cross-dataset evaluation for both tasks. Our method is beneficial for in-domain and out-of-domain datasets when we have access to a small set of samples with many annotations per instance ($\sim 100$) and for out-of-domain datasets when we have access to individual annotator samples ($> 2000$). Our work restates the potential of using selective prediction for subjective tasks as a research direction.

## 2 Soft Labels for Hate Speech Detection

Our method experiments with using soft labels as a candidate to better calibrate models for subjective tasks. Soft labels—where the model directly learns the human label distribution—have been used extensively in prior work on subjective task prediction (Jamison & Gurevych, 2015; Uma et al., 2020; 2021b). Based on this, we ask whether soft labels could also help improve model calibration, compared to hard labels. Moreover, these calibrated confidence estimates could be used to make the model abstain from any decision (Chow, 1957; Geifman & El-Yaniv, 2017) when there is greater annotator disagreement.

### 2.1 Background

A majority vote over the annotation count of a sample $x$: $y_h = \arg\max_{l \in \mathcal{y}_x}(freq(l))$, results in *hard labels* ($Y_h$) as the ground truth, used in training with the Cross-Entropy loss (**$CE_{hard}$**). However, for subjective tasks, majority voting might result in loss of vital information about the annotator disagreement and unjustified high model confidences, necessitating *soft labels*. Soft labels ($Y_s$) are the probability distribution over the classes of the annotator judgments for a sample. This can be done by e.g. normalizing the votes for each class or taking the softmax (Uma et al., 2020) when the number of annotators per sample is low. We illustrate the distinction between hard and soft labels in Figure 1.

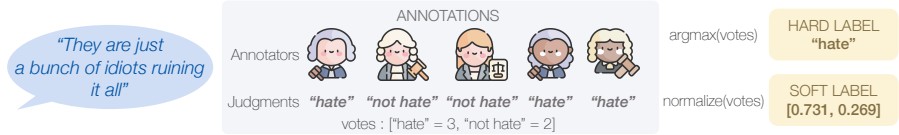

Figure 1: Example showing the difference between a hard and a soft label.

**Training.** Given a model $f$ that takes as input text $x$ and predicts the soft label $y_s$, the objective involves minimizing the distance between two probability distributions: the model distribution output $\hat{y}_s$ and the soft label $y_s$. This has been done by extending the Cross-Entropy Loss to measuring how distant $\hat{y}_s$ and $y_s$ are from each other (Peterson et al., 2019; Uma et al., 2020), using KL Divergence, Jensen-Shannon Divergence (***JSD***; the symmetric version of KL Divergence) or Mean Squared Error (Uma et al., 2021b).

**Evaluating.** Unlike *hard* evaluation metrics such as accuracy or F1-score, evaluating soft label prediction requires distance-specific metrics (Basile et al., 2021). Uma et al. (2021b) suggest using the soft losses mentioned as *soft* metrics. Baan et al. (2022) suggest using the Total Variation Distance (**TVD**).

## 2.2 Experimental Setup

We first investigate the utility of soft labels for the task of hate speech detection, where we have few annotations per example. If samples with more disagreement would then get lower confidences,[1] we can then use these confidences as a way to prevent a model from predicting when there is no clear consensus.

**Data.** We combine two widely used datasets: HateXplain (HX; Mathew et al., 2021) and the Measuring Hate Speech Corpus (MHSC; Sachdeva et al., 2022) for our experiments. Both provide the raw annotations for each sample. MHSC has one to five annotations per instance and a binary hate speech category. The HX dataset has three annotations per instance and consists of three classes: *hate speech, offensive, neutral*. In Appendix H we apply CrowdTruth (Dumitrache et al., 2018) as a dataset analysis. To match the datasets, we merge the *hate speech* and *offensive* classes in HX, essentially reducing the task to identifying offensive labels. The hard labels are derived by majority voting and the soft labels by using `softmax` over the annotator votes per class (in line with Figure 1), due to the low annotations per instance. We split the dataset into train (41832 samples), validation (5230 samples), and test (5230 samples).

**Model.** We use RoBERTa-large (Liu et al., 2020)[2] initialized with five different random seeds. We follow the original hyperparameters as described in Appendix C. Model selection is based on $JSD$ when training with $JSD$ loss and macro F1 when training with $CE_{hard}$.

**Metrics.** We compare the performance of models trained with $CE_{hard}$ and $JSD$ using *macro F1*, soft metrics ($JSD$ and $TVD$), and confidence calibration metrics. Soft labels could provide better confidence estimates of the model's empirical performance. To measure this, we use Expected Calibration Error (ECE; Naeini et al., 2015) and Brier Score (BS; Brier, 1950). For ECE, the confidence scores of the model are binned and for each bin, the accuracy is calculated. The average difference between the calculated and expected accuracy is taken over all bins. This metric is sensitive to the bin size. BS measures the mean squared error between the model confidence and its predictions. In NLP, only the calibration of the top confidence score is usually measured. This is a much weaker notion as it ignores the other dimensions (Vaicenavicius et al., 2019).

## 2.3 Soft vs. Hard Labels Results

Figure 2 shows the results[3]. As expected, using soft labels with few annotations per instance leads to (slightly) better performance on soft and calibration metrics ($JSD, TVD, ECE$, and $BS$). There is not much difference between the Macro F1-scores for the hard and soft models, in line with previous research.

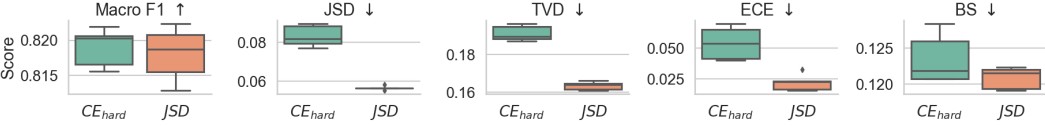

Figure 2: Results of training RoBERTa-large using hard (green, left) vs. soft labels (orange, right). Arrows indicate if a higher or lower value is better.

**When looking at the confidence distributions for both the hard and soft loss models, there is not a lot of difference in general.** In Figure 3, we visualize the confidence distributions

---

[1]With model confidence for both hard and soft labels as output we mean the softmax scores outputted by the model. In this paper, we do not consider model confidence in the output distribution it predicts as a soft label.

[2]Using the `HuggingFace Transformers` library (Wolf et al., 2019).

[3]In Appendix F we show results when training with other soft losses.

on samples for two types of agreement: **perfect agreement**, a sample where all annotators agree with each other on the assigned label, and **disagreement**: a sample where at least one annotator assigns a different label than the rest. In general, when training with $CE_{hard}$ (Figure 3a), we see a strongly unimodal confidence distribution for samples with perfect agreement. The skew on disagreement samples is less but still similar to perfect agreement. For the soft loss (Figure 3b), there is more mass distributed around lower confidences, especially for disagreement samples. Ideally, we would want fewer high confidences. When we disentangle correct and incorrect predictions for both perfect agreement (PA) and disagreement (DIS) samples, we see that with the hard loss (Figure 3c), regardless of correctness, for both agreement types most of the density is around the higher confidences. For the soft loss (Figure 3d), the confidences are more dispersed, especially for perfect agreement. Correct samples with perfect agreement are denser around higher confidences and incorrect perfect agreement samples are denser around lower confidences. This difference in distribution based on correctness for perfect agreement is not as notable for disagreement samples, which is more spread out but still has more mass in higher regions.

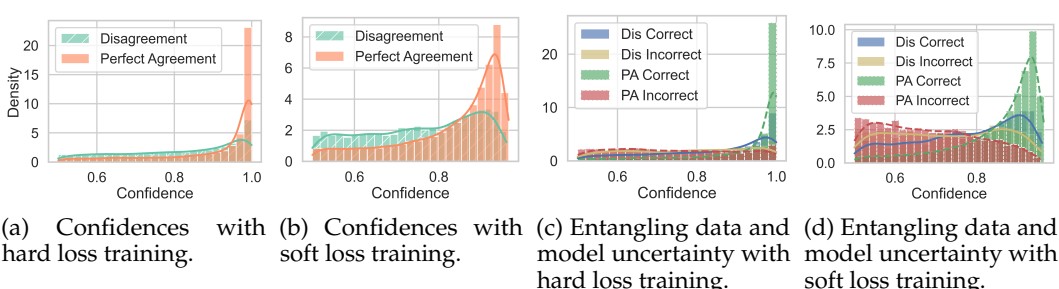

(a) Confidences with hard loss training.

(b) Confidences with soft loss training.

(c) Entangling data and model uncertainty with hard loss training.

(d) Entangling data and model uncertainty with soft loss training.

Figure 3: Visualizations of the confidence distributions for both perfect agreement and disagreement samples separately in the test set. Figure (a) shows the distribution for models trained with $CE_{hard}$ and Figure (b) shows it for models trained with $JSD$. In Figures (c) - (d), we visualize the confidence distributions for four different scenarios: {Perfect Agreement, Disagreement} × {Correct, Incorrect}.

We want disagreement samples to have generally lower confidence scores but our analysis shows that this is generally not the case. Our notion of disagreement in these experiments is based on 3 - 5 annotators, which can be limited in learning the crowd distribution. This motivates us to find a different method to make models more aware of human subjectivity.

## 3 Selective Prediction

The selective prediction framework (Geifman & El-Yaniv, 2017; Chow, 1957) aims to reduce errors made by a model by giving it the option to *abstain*. When dealing with subjective tasks, we argue that this is an attractive setup to align model output accordingly; abstain when dealing with highly subjective input and let predictions with more agreement, hence clear(er) correct answers, through. This setup consists of a base model $f(x)$ that outputs a softmax confidence distribution $\hat{y}$ that we want to *calibrate* accordingly and a decision function $h(x)$ on top, which determines if $f(x)$ predicts or abstains. Existing decision functions can either be confidence-based or a separate calibrator model:

**MaxProb** This method makes use of confidence thresholding. The highest confidence $\max(\hat{y})$ is compared to a threshold $t$. If the confidence is higher than $t$, the model predicts, but if the confidence is lower than $t$, the model abstains. We can apply MaxProb to $f(x)$ trained with either a hard loss with majority vote labels ($\hat{y}_{maj}$) or a soft loss ($\hat{y}_s$).

**Kamath et al. (2020)** Originally, this setup uses a separate calibrator that predicts if the base model is correct or not for the task of Question Answering in the case of domain shift where the base model is only trained on in-domain data. The calibrator, a random forest

classifier, is trained on both in and out-of-domain data. We draw an analogy between their in- & out-of-domain setup with perfect and disagreement in our task. In our case, the base model is only trained on samples with perfect agreement, $m_{PA}$, and the calibrator $h$ is trained on both disagreement and perfect agreement samples and outputs $h(x) = z$; if $m_{PA}$ is correct or not. If $z = 1$, only then does the prediction go through, otherwise, the model abstains. We train the calibrators with a random forest from XGBoost (Chen & Guestrin, 2016). We use the following input features: hidden state of [CLS] of base model's last layer (Corbière et al., 2019; Zhang et al., 2021; Zhang & Choi, 2023), base model's softmax probabilities, and n-grams from the calibration data.

## 4 CROWD-CALIBRATOR

Most of the calibration research has focused on the alignment between a model's confidence and its predictive performance. However, for subjective tasks, we want this to extend to reflecting human uncertainty as well, i.e. we want a *soft* calibration approach. Instead of focusing on the *correctness* of the model, we ask if *the softmax confidence distribution of a model is close to the crowd distribution for a given sample*.

We propose **CROWD-CALIBRATOR** (Figure 4), a *soft* calibration approach where we calibrate a model according to how close its confidence is to the crowd opinion. If the model confidence is far off from the crowd distribution, we want the model to *abstain*, i.e. *not* make a prediction. If the model confidence is close to the crowd distribution, we let the model predict. When the base model is overconfident on a sample where humans tend to disagree and thus there is no clear label, we can prevent the model from making a prediction.

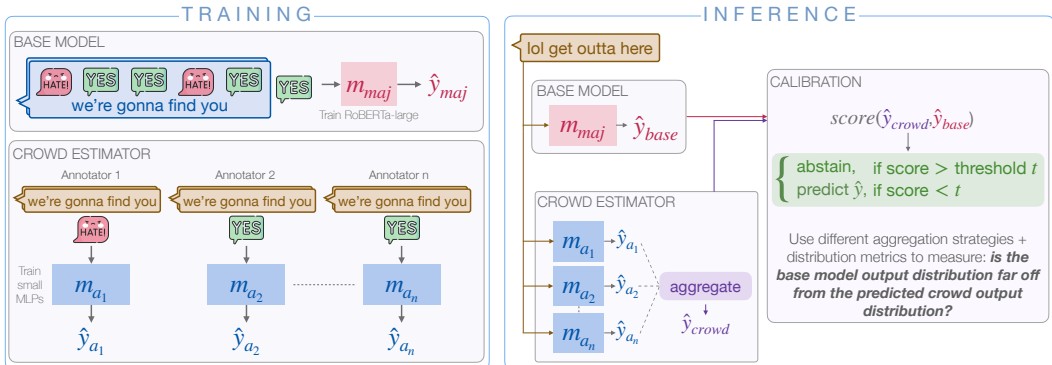

Figure 4: **CROWD-CALIBRATOR**: our proposed soft calibration approach where we calibrate a model according to human subjectivity. We only let the model predict if there is high agreement and thus a clear(er) correct answer. For NLI we directly train our crowd estimator to predict the crowd distribution as we only have access to the label distribution and not individual annotators.

### 4.1 Pipeline

Our setup consists of three components: the **base model** that we are calibrating, the **crowd estimator** that estimates the crowd judgment, and the soft **calibration**. Our base model $m_{maj}$, trained to predict the conventional majority vote label, is the model we want to calibrate. This is the same model trained with the hard loss in Section 2, outputting the softmax confidence distribution $\hat{y}_{maj}$ (also marked as $\hat{y}_{base}$ in Figure 4 *inference*).

#### 4.1.1 CROWD ESTIMATOR

With our **CROWD ESTIMATOR** we want to predict the human annotation distribution considering we want to calibrate our base model $m_{maj}$ according to human judgments.

However, we do not have this information available at test time. Our goal with the **CROWD ESTIMATOR**[4] is to estimate the crowd (dis)agreement of a given sample $x$.

The hate speech datasets that we have used until now are limited in terms of *modeling* the crowd. While there might be many annotators, the number of coders per instance is low (1 - 5) and many annotators have only seen a handful of samples. As there are no hate speech datasets with more than 5 annotations, we need an alternative to estimating the crowd. We *do* have access to hate speech datasets where a decent amount of annotators have seen a handful of samples: the Gab Hate Corpus (GHC; Kennedy et al., 2022) and the Dynamically Generated Hate Speech Dataset (DGHS; Vidgen et al., 2021). We select individual annotators that have annotated more than 2000 samples and train different individual MLPs $m_{a_i}$ (single layer, hidden size 512). We use the hidden state of [CLS] of base model's last layer as the input feature to predict if the annotator would perceive something as hate speech. To combine the individual annotator predictions into one crowd distribution $\hat{y}_{crowd}$, we employ three strategies: taking the individual predicted labels and creating a distribution over them with softmax (**Label Dist**), averaging over the confidence scores (**Avg. Conf.**) of the MLPs, and **Weighted Scoring** (Equation 1) where we take the weighted (fraction of voters $r_c$) distance score between the average confidence distribution of voters ($\mathcal{M}_c$) for a particular class $c$ and sum over all classes $C$.

$$WS = \sum_{c \in C} r_c \cdot score \left[ \left( \frac{1}{|\mathcal{M}_c|} \sum_{a \in \mathcal{M}_c} \hat{y}_a \right), \hat{y}_{maj} \right] \tag{1}$$

For NLI, we have enough instance-level annotations with ChaosNLI (Nie et al., 2020b). This dataset consists of 4500 re-annotated samples from SNLI (Bowman et al., 2015), MNLI (Williams et al., 2018), and $\alpha$NLI (Bhagavatula et al., 2019)[5]. Each sample gets 100 annotations, which makes it ideal to train our **CROWD ESTIMATOR** on. Our base model, RoBERTa-large, is trained on SNLI and MNLI with the majority vote. Instead of training separate annotator classifiers, we train a small MLP regressor (2-layered MLP with hidden sizes of 100 and the hidden state of [CLS] of the base model's last layer as input feature) that outputs soft labels directly. The further process is the same as shown in Figure 4.

### 4.1.2 Calibration

Now that we have both $\hat{y}_{maj}$ and $\hat{y}_{crowd}$, we want to measure their distance. Thus, we experiment with different distance metrics (KL divergence (KL), JSD, and TVD; see Appendix D for formulae). If the model's output distribution is far off from the human probability distribution, the model will abstain, otherwise the model's prediction, argmax($\hat{y}_{maj}$) will go through. We also experiment with adding the entropy (+E) of $\hat{y}_{maj}$ to the entire score to remove instances where both $\hat{y}_{crowd}$ and $\hat{y}_{maj}$ are close to each other but have low confidence.[6]

### 4.2 Metrics

We evaluate our *soft* calibrator through the lens of how well it abstains (does the calibrator let the correct samples through)[7] and how conservative it is (does it let through sufficient samples to achieve good performance).

**Cov@Acc=** Since the model abstains, we focus on **coverage**: the fraction of samples the base model predicts. The coverage changes as we adjust our threshold for the distance score or model confidence. By increasing the confidence or reducing the distance score, the coverage decreases and we expect accuracy to increase. Hence we compare coverage at a fixed accuracy value to evaluate how conservative a technique is at that point.

---

[4]Our Crowd Estimator MLPs (both tasks) are trained with scikit-learn (Pedregosa et al., 2011).

[5]We filter out samples from $\alpha$NLI as it has different classes. See Appendix C for details.

[6]Adding the entropy to the KL divergence does not result in canceling out the entropy already present in the equation and reducing down to the cross entropy. We write out the proof in Appendix B.

[7]Note that using the notion of correctness does not mean that we assume a single ground truth, we view it as observing a single sample of the underlying crowd distribution.

**AUC and AUBS**   We yield corresponding accuracies for each coverage when thresholding. Following Geifman & El-Yaniv (2017), we calculate the area under the accuracy-coverage curve (**AUC**) to quantify how much the calibrator abstains and how correctly it abstains over the entire trajectory. Similarly, to measure if a calibration method improves calibration as coverage decreases, we compute the area under the coverage-BS curve (**AUBS**).

**AUROC**   Following Shrivastava et al. (2023), we want to know if the thresholding or confidence has good differentiating power between correct and incorrect predictions, which is why we plot the fraction of wrong samples let through against the fraction of correct samples let through and calculate the area under this curve.

## 5   Results

We discuss the results of our soft calibrator obtained for both *hate speech* and *NLI*[8]. We also explore the soft calibrator's capabilities on other unseen datasets for both tasks. For all results, we highlight the  best performance in blue  and the  worst performance in red . All results are averaged over five seeds.[9]

### 5.1   Hate Speech: Calibrating from Individual Annotators

We present the results obtained in Table 1, where we use a combined test set of perfect agreement and disagreement with 3200 samples. We also experimented with GPT-4 but did not yield competitive results, which we present in Appendix A. To verify our hate speech results, we also experiment with *Online Misogyny* in Appendix E, obtaining similar results.

|  |  | **Cov@Acc=** $\uparrow$ | | | | | |
|---|---|---|---|---|---|---|---|
|  |  | **0.85** | **0.90** | **0.95** | **AUC** $\uparrow$ | **AUROC** $\uparrow$ | **AUBS** $\downarrow$ |
| **MaxProb** | $\hat{y}_{PA}$ | 0.8733 | 0.6890 | 0.4422 | 0.9277 | 0.7775 | 0.0638 |
|  | $\hat{y}_{maj}$ | 0.8757 | 0.6934 | **0.4839** | **0.9302** | 0.7807 | 0.0592 |
|  | $\hat{y}_{soft}$ | **0.8799** | 0.6772 | 0.3241 | 0.9184 | 0.7574 | 0.0692 |
|  | $\hat{y}_{maj}$ - TS | 0.8751 | 0.6937 | 0.4834 | 0.9301 | 0.7807 | **0.0567** |
|  | Kamath et al. (2020) | 0.8683 | 0.6866 | 0.4775 | **0.9302** | **0.7830** | 0.0597 |
| **Ours** | Label Dist. - JSD+E | 0.8571 | 0.6717 | 0.4663 | 0.9281 | 0.7713 | 0.0601 |
|  | Label Dist. - TVD+E | 0.8519 | 0.6712 | 0.4624 | 0.9270 | 0.7663 | 0.0605 |
|  | Avg. Conf. - JSD+E | 0.8728 | **0.6975** | 0.4640 | 0.9283 | 0.7780 | 0.0609 |
|  | Weighted Scoring - JSD+E | 0.8666 | 0.6959 | 0.4507 | 0.9275 | 0.7763 | 0.0616 |
|  | GHC - Avg. Conf. - JSD+E | 0.8668 | 0.6974 | 0.4721 | 0.9288 | 0.7790 | 0.0602 |
|  | GHC - Label Dist. - JSD+E | 0.8606 | 0.6854 | 0.4777 | 0.9290 | 0.7754 | 0.0591 |

Table 1: Calibration results on the combined HX + MHSC hate speech test set for calibration. The upper part shows the results using MaxProb with RoBERTa-large models trained on different types of data ($\hat{y}_{maj}$, $\hat{y}_{PA}$), soft loss ($\hat{y}_{soft}$), or using better-calibrated confidences with Temperature Scaling (TS) (Guo et al., 2017). We also show MaxProb when training with only perfect agreement samples ($\hat{y}_{PA}$) for completeness with Kamath et al. (2020)'s base model. The lower part shows the best-performing results using our proposed CROWD-CALIBRATOR with RoBERTa-large as a base model and combining individual annotator models for the CROWD-ESTIMATOR.

We see that $\hat{y}_{maj}$ performs the best in general. The Kamath et al. (2020) baseline performs closely, with the same AUC and slightly better AUROC ($\Delta$0.0023). However, Kamath et al. (2020) has a more complex setup in front of MaxProb. MaxProb on soft labels $\hat{y}_{soft}$ performs worse than $\hat{y}_{maj}$ in terms of AUC and AUROC, highlighting the findings discussed in Section 2.3. For **CROWD-CALIBRATOR**, averaging over the annotator confidences and

---

[8]Code released here: https://github.com/urjakh/crowd-calibrator
[9]For the variation and standard deviation in change of performance with CROWD-CALIBRATOR compared to the baseline, see Appendix G.

using JSD as a distance metric with entropy yields the best results. It also gets the highest Cov@Acc=90. The best AUC is 0.9290 (-Δ0.12%) and AUROC is 0.7790 (-Δ0.17%).

We present results obtained on unseen datasets in Table 2. We apply the best-performing aggregation strategies to the Davidson (5000 random samples) (Davidson et al., 2017), Founta (2500 negative and 2500 positive random sampled) (Founta et al., 2018), and full HateCheck (Röttger et al., 2021) datasets.

| | DAVIDSON | | | FOUNTA | | | HATECHECK | | |
|---|---|---|---|---|---|---|---|---|---|
| | AUC ↑ | AUROC ↑ | AUBS ↓ | AUC ↑ | AUROC ↑ | AUBS ↓ | AUC ↑ | AUROC ↑ | AUBS ↓ |
| MaxProb $\hat{y}_{maj}$ | 0.6980 | 0.5692 | 0.3006 | 0.7695 | 0.6543 | 0.2195 | 0.5702 | 0.5557 | 0.3973 |
| Kamath et al. (2020) | 0.6723 | 0.5368 | 0.2691 | 0.7696 | 0.6457 | 0.2181 | 0.5294 | 0.6066 | 0.4412 |
| Avg. Conf. - TVD+E** | **0.7197*** | **0.6171*** | **0.2269** | 0.7890* | 0.6622 | 0.1981* | **0.7256*** | 0.8248* | **0.2502*** |
| Weighted Score - JSD+E** | 0.7109* | 0.5962* | 0.2388 | **0.7905*** | **0.6643*** | **0.1964*** | 0.6870* | 0.7477* | 0.2895* |
| DGHS - Label Dist. - TVD+E | 0.7144* | 0.6088* | 0.2310 | 0.7756 | 0.6575 | 0.2116* | 0.7172* | **0.8303*** | 0.2619* |

Table 2: Calibration results on unseen hate speech datasets: Davidson, Founta, and Hate-Check. We compare our CROWD-CALIBRATOR with the best-performing baselines. We show the results for the *Avg. Conf. - TVD+E* and *Weighted Score - JSD+E* aggregation strategies using DGHS annotators for Davidson and HateCheck and GHC annotators for Founta as these sets of annotators give the best results. *paired t-test with *p*-value $< 0.05$.

Here we see that our **CROWD-CALIBRATOR** can outperform both MaxProb and Kamath et al. (2020) on other datasets. In all cases, the best-performing method is a variant of our proposed setup. For Davidson and HateCheck, we show the results when only using DGHS annotators, and for Founta when only using GHC annotators (indicated with a **) as these give the best results. This reveals the sensitivity of our method to the individual annotators on which it is training. For the original test set, the Gab annotators were more useful instead.

## 5.2 NLI: Calibrating from Crowd Distributions

We show the results obtained on the test of ChaosNLI (312 samples) in Table 3. We only compare with MaxProb since Kamath et al. (2020) has a complex setup but, compared to MaxProb, yields comparable or worse results for hate speech. We see that with TVD and entropy, we can beat the baseline with an increase of 4.65% in AUC and 8.37% in AUROC.

| | cov@acc=0.8 ↑ | cov@acc=0.9 ↑ | AUC ↑ | AUROC ↑ | AUBS ↓ |
|---|---|---|---|---|---|
| MaxProb $\hat{y}_{maj}$ | 0.6244 | 0.0846 | 0.8114 | 0.6720 | 0.1147 |
| KL | 0.6167 | 0.2455* | 0.8226 | 0.6774 | 0.0881* |
| KL + E | 0.7276* | **0.3603*** | 0.8540* | 0.7400* | 0.0846* |
| TVD | 0.7128* | 0.3545* | 0.8468* | 0.7325* | **0.0802*** |
| TVD+E | **0.7596*** | 0.3558* | **0.8579*** | **0.7557*** | 0.0841* |
| JSD | 0.6833 | 0.2609* | 0.8304* | 0.7027* | 0.0862* |
| JSD+E | 0.7045* | 0.2801* | 0.8435* | 0.7262* | 0.0925* |

Table 3: Calibration results on the test set of ChaosNLI. We compare the best-performing selective prediction baseline (MaxProb $\hat{y}_{maj}$) based on performance on the hate speech sets. For CROWD-CALIBRATOR, we show the results for different distance metric variations and include entropy into the score or not. *paired t-test with *p*-value $< 0.05$.

We also apply our method to other datasets where it beats the baseline: the test set of the ANLI (Nie et al., 2020a) (3200 samples) dataset and the test set of WANLI (5000 samples) (Liu et al., 2022). We show these results in Table 4. For ANLI, adding the entropy decreases performance but MaxProb still performs the worst out of all. For WANLI too, our soft calibration method with JSD and entropy gets the highest performance and beats MaxProb. The specific strength of the calibrator seems to be that it maintains higher coverage when targeting higher accuracy.

| | ANLI | | | WANLI | | | | | |
| --- | --- | --- | --- | --- | --- | --- | --- | --- | --- |
| | | | | cov@acc= ↑ | | | | | |
| | AUC ↑ | AUROC ↑ | AUBS ↓ | 0.7 | 0.75 | 0.8 | AUC ↑ | AUROC ↑ | AUBS ↓ |
| MaxProb $\hat{y}_{maj}$ | 0.2809 | 0.4503 | 0.4300 | 0.6446 | 0.3807 | 0.1099 | 0.7259 | 0.6288 | 0.1711 |
| KL | **0.3345*** | **0.5285*** | **0.3400*** | | | | | | |
| KL+E | 0.3026* | 0.4877* | 0.4037* | 0.6177 | 0.3364 | 0.1450 | 0.7276 | 0.6217 | 0.1658 |
| TVD | 0.3266* | 0.5185* | 0.3583* | | | | | | |
| TVD+E | 0.3011* | 0.4810* | 0.4072* | 0.6398 | 0.3623 | 0.1424 | 0.7309* | 0.6290 | 0.1646 |
| JSD | 0.3336* | 0.5248* | 0.3461* | | | | | | |
| JSD+E | 0.2930* | 0.4674* | 0.4176* | **0.6450** | **0.3989** | **0.1878** | **0.7350*** | **0.6346** | **0.1638*** |

Table 4: Calibration results on other unseen NLI datasets. *paired t-test with $p$-value $< 0.05$

### 5.3 Qualitative Examples ( ⚠ Offensive Content )

Though a full manual qualitative analysis is out of the scope of this paper, we illustrate CROWD-CALIBRATOR with some examples of its output in Table 5. We selected examples that either have very small scores (obvious pass-throughs) or large ones (obvious abstains) according to CROWD-CALIBRATOR so that they are representative of various thresholds. As such, for hate speech, the shown examples are obvious cases of the ground truth when let through but rather debatable when abstained. For NLI, a misclassification is prevented. These examples indicate that our approach may be useful for manual analysis of the data, e.g. retrieving cases of agreement between the base model and CROWD-CALIBRATOR but different ground truth can be a quick tool to find (some) annotation mistakes.

| | $m_{maj}$ | CROWD-CALIBRATOR | Ground Truth |
| --- | --- | --- | --- |
| I will s*ck yo d*ck and call you a f*gg*t | hate | ✓ | hate |
| This is amazing! Kudos for that women for being so strong | non-hate | ✓ | non-hate |
| Imagine the kid was black in a white class, .., liberal lunatics would go to war with the white race | hate | ✗ | non-hate |
| \<user\> Men r trash tho right | non-hate | ✗ | hate |
| a woman walks with her big purse. a beautiful female passing by crowd | neutral | ✓ | neutral |
| bicycles stationed while a group of people socialize. people talk near a bunch of parked motorcycles. | entailment | ✗ | contradiction |

Table 5: Examples of what CROWD-CALIBRATOR abstains (✗) on and lets the base model $m_{maj}$ predict (✓) for both hate speech and NLI (premise, hypothesis).

## 6 Related Work

**Learning from disagreement.** Recent research has advocated incorporating the notion of subjectivity into our models and NLP pipeline (Aroyo & Welty, 2015; Cabitza et al., 2023; Plank, 2022; Plank et al., 2014; Nie et al., 2020b; Chklovski & Mihalcea, 2003). A direct way is to move away from *hard* labels to *soft* labels (Uma et al., 2021a; Peterson et al., 2019; Wu et al., 2023; Jamison & Gurevych, 2015; Fornaciari et al., 2021). Such labels can also be used to model an annotator's uncertainty for a label's distribution (Liu et al., 2019). Other approaches focusing on annotation disagreement range from a probabilistic setup (Raykar et al., 2010) to modeling the annotators themselves (Rodrigues & Pereira, 2018; Davani et al., 2022; Guan et al., 2018). Zhou et al. (2022) explore distribution estimation methods. Utilizing disagreement for hate speech detection systems has had limited attention. Leonardelli et al. (2021) investigate the use of different data splits based on (dis)agreement for offensive language and Al Kuwatly et al. (2020) group annotators based on their demographic to investigate their biases for hate speech classification. Davani et al. (2022) show the benefit of using individual annotator information but require data where individual annotators have labeled many samples. This is something that many datasets including the ones we used in the paper—the HateXplain, Measuring Hate Speech Corpus, and NLI datasets—do not have. Hence, we could not empirically compare to their approach.

**Calibration.** Investigating if a model is well-calibrated has been gaining traction (Desai & Durrett, 2020; Jiang et al., 2021b; Corbière et al., 2019; Baan et al., 2022; Ulmer et al., 2022; Nalisnick et al., 2018; Hendrycks & Gimpel, 2016). Guo et al. (2017) show how modern neural networks are miscalibrated. Recalibration can be done post-hoc (Guo et al., 2017; Gal & Ghahramani, 2016; Jiang et al., 2021b) or through a hybrid human-model approach (Kerrigan et al., 2021), amongst others. Efforts are being made to understand the calibration of LLMs, through their verbal (Tian et al., 2023; Krause et al., 2023) and probabilistic confidence (Shrivastava et al., 2023). Shrivastava et al. (2023) show how using a mixture of linguistic and probabilistic confidences leads to better calibration. Vidgen et al. (2020) use a Bayesian approach to recalibrate models for abusive language and show that uncalibrated models are not in line with annotators.

**Selective prediction.** Selective prediction/learning to abstain is a relaxation of *learning to defer* (Madras et al., 2018): when a model determines when to defer to an expert by modeling that expert's knowledge. Learning to abstain/reject, on the other hand, does not model the expert, implicitly assuming a fallback mechanism with uniformly better accuracy than the rejection threshold. Learning to defer has been extended to multiple experts (Verma et al., 2023). In NLP (Xin et al., 2021), there has been a lot of focus on selective prediction for Question Answering (Kamath et al., 2020; Rodriguez et al., 2019; Zhang et al., 2021; Zhang & Choi, 2023; Varshney & Baral, 2023; Cole et al., 2023; Yoshikawa & Okazaki, 2023).

## 7  Discussion and Conclusion

We propose CROWD-CALIBRATOR, a soft calibration method for subjective tasks that refrains from predicting if its confidence is far from the crowd label distribution. We show the utility of our method by applying it to two complementary data scenarios: the real-world scenario with only access to a large number of individual annotations and the ideal scenario with a large number of sample annotations. Our experiments show that CROWD-CALIBRATOR outperforms the MaxProb baseline for the task of NLI, both on the respective test set (albeit small) and two other datasets. For hate speech, our method is competitive with the baselines on the respective test set and outperforms baselines on unseen datasets.

Our proposed setup clearly works with fewer samples but requires a high amount of sample annotations ($\sim 100$) to estimate the crowd distribution. This is less the case when we estimate the crowd distribution from individual annotators. Yet the number of annotators varied across tasks, with fewer annotators available for hate speech (24) than for NLI (100). Therefore, it is clearly beneficial to have a high amount of instance-level annotations, as also seen in Gruber et al. (2024). In future work, investigating the number of annotators and its effect on our method poses an obvious and interesting direction. Thus applying our method to a novel task means that we need at least a small amount of multiply annotated data reflecting different annotator judgments. As such data is not easily available for many subjective NLP tasks, this is a limitation that needs to be considered.

**Takeaways.** While our method shows sensitivity to the number of annotations, in both data scenarios we see notable results for unseen data. Our method offers flexibility in the choice of the distance metric and aggregation strategy. Which combination works best is largely dataset-dependent. The various settings we experiment with are not exhaustive options and can easily be replaced by other metrics or strategies. Generally, JSD and TVD gave the best results in combination with aggregating through label distribution or averaged confidences. In all cases, it is beneficial to add entropy to the score of the metric as it prevents the model from making decisions when both the model and crowds are uncertain.

In line with other work advocating for more subjectivity-informed model behavior, we hope our findings encourage more research in the direction of subjectivity-based selective prediction and better design of datasets and model setup for subjective tasks. If we really want to understand the crowd and create more human-aligned applications, we need many more instance-level annotations from a diverse set of annotators with a high individual annotation count.

## Acknowledgments

We thank Rajeev Verma for his helpful feedback on the draft version. We are also grateful to the anonymous reviewers for their valuable feedback that helped improve the paper. The figures and affiliation emojis have been designed using resources from *flaticon.com*; the tulip is created by *smashicons*, crab by *mihimihi*, and the rest of the emojis by *freepik*. This research was (partially) funded by the Hybrid Intelligence Center, a 10-year programme funded by the Dutch Ministry of Education, Culture and Science through the Netherlands Organisation for Scientific Research, as well as by awards from the Allen Institute for AI and Intel Labs.

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

## A    Results with LLMs

We use GPT-4 (`gpt-4-1106-preview`) (Achiam et al., 2023) to investigate if open-source LLMs are useful in detecting hate speech and if their linguistic and model confidence is aligned with human subjectivity. Following Tian et al. (2023), we prompt the model to detect if a given sample is hate speech and if it can express its verbal confidence (**GPT-4 V**) through a score between 0 and 1. We also take the log probabilities (**GPT-4 L**) of the label the LLM predicts.

We use the following prompt, following Tian et al. (2023):

---

**GPT-4 Prompt**

```
" Classify with one of the following labels:  (a) not hate speech,  (b)
hate speech, and the probability that it is correct (0.0 to 1.0) for the
following sentence. Give ONLY the label and probability, no other words or
explanation.

Follow this format:

Label: <most likely label, either a or b, as short as possible; not a
complete sentence, just the label!> Probability: <the probability between
0.0 and 1.0 that your guess is correct, without any extra commentary
whatsoever; just the probability!>
The sentence is: """{sentence}""". "
```

---

We show the results in Table 6, which clearly show that GPT-4, with both sources of confidence, is not able to get close to the simplest baseline of MaxProb $\hat{y}_{maj}$, not even reaching an accuracy of 0.95 when using verbal confidences.

| | Cov@Acc= ↑ | | | AUC ↑ | AUBS ↓ |
|---|---|---|---|---|---|
| | **0.85** | **0.90** | **0.95** | | |
| $\hat{y}_{maj}$ | 0.8757 | 0.6934 | **0.4839** | **0.9302** | 0.0592 |
| GPT-4 V | 0.4936 | 0.2993 | - | 0.8320 | 0.1390 |
| GPT-4 L | 0.5870 | 0.4516 | 0.2512 | 0.8756 | 0.0875 |

Table 6: Calibration results on the combined HX + MHSC hate speech test set for calibration. The upper part shows the results using MaxProb with RoBERTa-large when training on the majority vote. We compare this to GPT-4 when using verbal confidence estimates and the actual log probs for the label.

We also attempted using LLAMA-2 7B Chat (Touvron et al., 2023) but due to the strict safety instructions, it refused to classify hate speech.

## B    Equation KL Divergence with Entropy

As a score to calibrate our model with, we initially used only the KL Divergence between the *crowd observation P* and the *model softmax distribution Q*: $KL(P||Q)$. This, however, does not take into account when both $P$ and $Q$ have low confidence. To filter out such cases, we add the entropy of $Q$: $H(Q)$. When both $P$ and $Q$ are close to each other and there is high confidence, the difference score will be small. If both are close to each other but there is low confidence, the difference score will still be larger due to the added entropy. We write out the entire equation below:

$$KL(P||Q) = \sum_{x \in X} P(x) log\left(\frac{P(x)}{Q(x)}\right)$$

$$H(Q) = -\sum_{x \in X} Q(x) \log(Q(x))$$

$$KL(P||Q) + H(Q) = \left[\sum_{x \in X} P(x) log\left(\frac{P(x)}{Q(x)}\right)\right] + \left[-\sum_{x \in X} Q(x) \log(Q(x))\right]$$

$$= \left[\sum_{x \in X} P(x)(\log(P(x)) - \log(Q(x)))\right] - \sum_{x \in X} Q(x) \log(Q(x))$$

$$= \sum_{x \in X} P(x) \log(P(x)) - P(x) \log(Q(x)) - \sum_{x \in X} Q(x) \log(Q(x))$$

$$= \sum_{x \in X} P(x) \log(P(x)) - P(x) \log(Q(x)) - Q(x) \log(Q(x))$$

## C  Training Details

For each experiment, we give the hyperparameters and the dataset sizes used. All of our results are aggregated over 5 random seeds and our experiments are done on a Titan 6000.

**RoBERTa-large on MHSC + HX (soft and hard)**    To train our models we follow the original hyperparameters. We train for 10 epochs, with a learning rate of $1e^{-5}$, weight decay of 0.1, and 6% warmup steps. We use a training batch size of 8. We train these models on 41832 samples and validate on 5230. The results shown in Section 2.3 are on the test set with 5230 samples.

**Kamath et al. (2020)**    We use the same original hyperparameters for the RoBERTa-large base model that is now only trained on 32316 perfect agreement samples and validated on 4039 perfect agreement samples from MHSC + HX. Our Random Forest calibrator, trained with `XGBoost` has a learning rate of 0.07, max depth of 5, and 500 parallel trees. The calibrator is trained on a mixture of perfect agreement samples (2020) and the rest disagreement samples to bring the total to around 7000 samples to balance according to correctness and agreement.

**Individual Hate Speech Annotators**    Each annotator dataset is split into 80% training, 10% validation, and 10% test. For each annotator, we train an MLP, that is a single layer MLP with a hidden size of 512, for the rest using the default parameters in `scikit-learn`.

**ChaosNLI**    ChaosNLI comes with 4500 samples from which we remove the $\alpha$NLI samples since those have different classes. We train on 2490 samples, validate on 311 samples, and test on 312 samples. We use a two-layer MLP regressor with hidden sizes of 100, for the rest using the default parameters in `scikit-learn`.

**Selective Prediction - MHSC + HX**    The selective prediction results for hate speech on MHSC + HX are done on 3200 samples with mixed perfect and disagreement.

## D  Distance Metrics Formulae

To calibrate our base model $m$ in a *soft* fashion, we need distance metrics to determine the proximity between the base model's predicted distribution $\hat{y}_{base}$ (also referred to as $\hat{y}_{maj}$ when the base model $m_{maj}$ is trained to predict the majority vote) and the crowd distribution $\hat{y}_{crowd}$. We experiment with the following widely-used distance metrics:

**KL DIVERGENCE (KL, Equation 2)**    measures the difference of one reference distribution from the other, however, it is not symmetric. If the other distribution is the reference

distribution, we will get a different output.

$$KL(\hat{y}_{base}, \hat{y}_{crowd}) = \hat{y}_{crowd} \cdot (\hat{y}_{crowd} - \log \hat{y}_{base}) \tag{2}$$

**JENSEN-SHANNON DIVERGENCE (JSD, Equation 3)** measures the distance between the two distributions and is a symmetrical version of the KL divergence.

$$JSD(\hat{y}_{base}, \hat{y}_{crowd}) = \frac{1}{2}KL\left(\frac{\hat{y}_{base} + \hat{y}_{crowd}}{2}, \hat{y}_{base}\right) + \frac{1}{2}KL\left(\frac{\hat{y}_{base} + \hat{y}_{crowd}}{2}, \hat{y}_{crowd}\right) \tag{3}$$

**TOTAL VARIATION DISTANCE (TVD, Equation 4)** is a way to measure the distance between two distributions by measuring the absolute difference in probabilities.

$$TVD(\hat{y}_{base}, \hat{y}_{crowd}) = \frac{1}{2}||\hat{y}_{base} - \hat{y}_{crowd}||_1 \tag{4}$$

## E  Results for Online Misogyny

To verify the results achieved on the hate speech dataset, we apply our CROWD-CALIBRATOR setup to a similar subjective task: *misogyny* detection. This dataset (Guest et al., 2021) has 6383 samples with 2-3 annotations for the majority of its samples (6259), with the rest having 4 - 16 annotations per sample. We train on 5096 samples and validate on 646 samples. This dataset is imbalanced, where most of the samples are non-misogynistic in comparison to misogynistic.

To train the CROWD ESTIMATOR, we use the EDOS dataset (Kirk et al., 2023), for sexism detection. We train individual MLPs just like for the hate speech dataset for annotators that have more than 3000 annotations. This gives us 14 individual annotators, which is less than what we had for the hate speech dataset.

Our results on the respective test set can be found in Table 7. Due to the imbalanced nature of the dataset, we balance out the test set to get a better view, where there are 72 misogynistic samples and 100 non-misogynistic.

| | cov@acc=0.85 ↑ | cov@acc=0.9 ↑ | cov@acc=0.95 ↑ | AUC ↑ | AUROC ↑ | AUBS ↓ |
|---|---|---|---|---|---|---|
| MaxProb $\hat{y}_{maj}$ | 0.8600 | 0.7087 | 0.4950 | 0.9244 | 0.7796 | 0.06672 |
| Label Dist. - TVD+E | 0.8675 | 0.7163 | 0.4975 | 0.9254 | 0.7870 | 0.06585 |
| Label Dist. - JSD+E | 0.8575 | 0.7075 | 0.4913 | 0.9249 | 0.7844 | 0.06604 |
| Label Dist. - KL+E | 0.8775 | 0.7163 | 0.4975 | 0.9252 | 0.7857 | 0.06587 |
| Avg. Conf. - TVD+E | 0.8729 | 0.7063 | 0.4650 | 0.9244 | 0.7871 | 0.06672 |

Table 7: Calibration results on the test set of the Online Misogyny dataset. We compare with the best-performing selective prediction baseline (MaxProb $\hat{y}_{maj}$). For CROWD-CALIBRATOR, we show the results for different distance metric variations and include entropy in the score.

We observe results similar to hate speech detection, with competitive performance.

## F  Comparing Distance Metrics for Soft Labels

We experiment with different loss functions that would minimize the difference between the annotator probability distribution $y_s$ and the model's output distribution $\hat{y}_s$: *Cross Entropy Loss* ($CE_{soft}$, Equation 5), *Jensen Shannon Divergence* (*JSD*, Equation 3), and *Mean Squared Error* (*MSE*, Equation 7). We refer to these losses as *soft losses*. For comparison to the conventional setup, we also use *Cross Entropy Loss* in the conventional setup and refer to this loss as a hard loss ($CE_{hard}$), where the objective is to predict the hard label $y_h$.

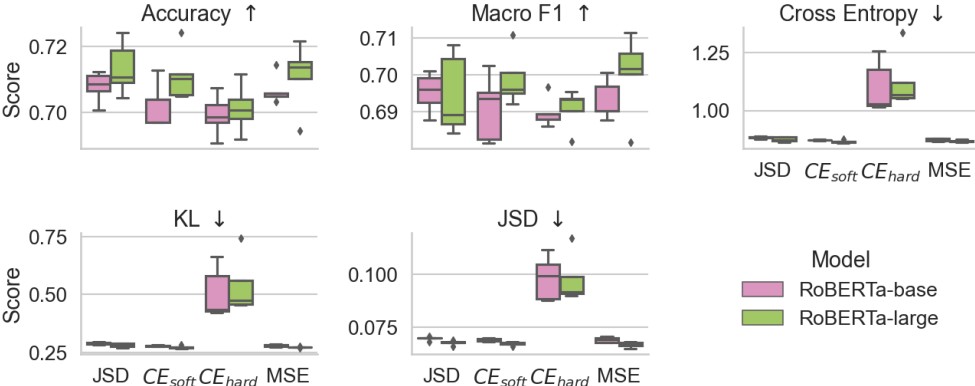

Figure 5: Results on the HateXplain dataset for both RoBERTa-base and RoBERTa-large when training with hard and soft losses (x-axis). Each plot is a different metric.

$$CE_{soft}(y_s, \hat{y}_s) = -\sum_{i=1}^{n} y_{s,i} \log(\hat{y}_{s,i}) \tag{5}$$

$$CE_{hard}(y_h, \hat{y}_h) = -\sum_{i=1}^{n} y_{h,i} \log(\hat{y}_{h,i}) \tag{6}$$

$$MSE(y_s, \hat{y}_s) = (y_s - \hat{y}_s)^2 \tag{7}$$

### F.1 Evaluation

To measure the performance and calibration of the model, we employ different metrics.

**Hard evaluation metrics.** We use these metrics to measure if the model gets its predictions right in the conventional way: *macro F1* and *accuracy*.

**Soft evaluation metrics.** To measure if the probabilities that the model outputs are similar to the annotator label distribution, we use $CE_{soft}$, *KL*, and *JSD*.

**Calibration metrics.** To measure if the confidence the model outputs reflects its empirical performance, we measure the *Expected Calibration Error (ECE)*.

### F.2 Results

**The hard and soft evaluation results** can be found in Figure 5. Here, we see that for all metrics, the hard loss is outperformed by other losses. This is even the case for hard metrics. The superior performance of the soft losses highlights how beneficial it is for models to learn the human label variation instead of the majority vote.

**To evaluate the losses in terms of calibration,** we show the Expected Calibration Error in Table 8. Generally, the hard loss has the worst (highest) *ECE* out of all and tends to be more overconfident. The soft losses have confidences closer to their empirical performance and tend to be rather underconfident. The *JSD* loss comes the closest to reflecting the actual performance of the model, having the lowest ECE. We therefore continue our experiments with only the *JSD* loss as a soft loss.

|  | $CE_{hard}$ | $CE_{soft}$ | JSD | MSE |
|---|---|---|---|---|
| **RoBERTa-base** | 9.67 | 4.55 | **3.21** | 4.65 |
| **RoBERTa-large** | 10.63 | 5.50 | **3.73** | 5.30 |

Table 8: ECE for different losses on the HateXplain dataset, averaged over five runs.

# G   Variation in Change of Performance with CROWD-CALIBRATOR

To showcase the standard deviation and variation of our results, we look at the improvement or deterioration for the metrics we apply when using CROWD-CALIBRATOR compared to MaxProb on the base model trained with majority labels. We take the score achieved with CROWD-CALIBRATOR and subtract the score of MaxProb $\hat{y}_{maj}$ to get the difference ($\Delta$) in performance, for each seed individually. After that, we take the average and calculate the standard deviation.

We show the mean variation in Tables 9 and 11 for hate speech and Tables 10 and 12 for NLI. The subscript is the standard deviation ($\pm$). A positive mean variation means that in general, our CROWD-CALIBRATOR performs better than the baseline MaxProb. A negative mean variation means generally worse performance.

For completeness, we also show the difference in performance between the two baselines of MaxProb and Kamath et al. (2020).

|  | Cov@Acc= ↑ | | | $\Delta$ AUC ↑ | $\Delta$ AUROC ↑ |
|---|---|---|---|---|---|
|  | $\Delta$ 0.85 | $\Delta$ 0.90 | $\Delta$ 0.95 | | |
| Kamath et al. (2020) | $-0.0074_{0.0197}$ | $-0.0068_{0.0169}$ | $-0.0064_{0.0282}$ | $0.0000_{0.0034}$ | $0.0022_{0.0086}$ |
| Label Dist. - JSD+E | $-0.0186_{0.0124}$ | $-0.0217_{0.0127}$ | $-0.0176_{0.0205}$ | $-0.0021_{0.0014}$ | $-0.0095_{0.0082}$ |
| Label Dist. - TVD+E | $-0.0238_{0.0188}$ | $-0.0222_{0.0117}$ | $-0.0215_{0.0085}$ | $-0.0031_{0.0015}$ | $-0.0145_{0.0066}$ |
| Avg. Conf. - JSD+E | $-0.0029_{0.0209}$ | $0.0041_{0.0230}$ | $-0.0199_{0.0288}$ | $-0.0019_{0.0033}$ | $-0.0027_{0.0087}$ |
| Weighted Scoring - JSD+E | $-0.0091_{0.0066}$ | $0.0025_{0.0091}$ | $-0.0332_{0.0180}$ | $-0.0027_{0.0013}$ | $-0.0044_{0.0029}$ |
| GHC - Avg. Conf. - JSD+E | $-0.0089_{0.0085}$ | $0.0040_{0.0085}$ | $-0.0118_{0.0324}$ | $-0.0014_{0.0009}$ | $-0.0017_{0.0058}$ |
| GHC - Label Dist. - JSD+E | $-0.0151_{0.0051}$ | $-0.0080_{0.0116}$ | $-0.0062_{0.0130}$ | $-0.0011_{0.0009}$ | $-0.0054_{0.0038}$ |

Table 9: Difference ($\Delta$) in increase (+) or decrease (-) of metric score when calibrating with CROWD-CALIBRATOR compared to MaxProb $\hat{y}_{maj}$ for the **in-domain hate speech test set**. We present the mean difference, with the standard deviation $\pm$ as subscript.

|  | $\Delta$ cov@acc=0.8 ↑ | $\Delta$ cov@acc=0.9 ↑ | $\Delta$ AUC ↑ | $\Delta$ AUROC ↑ |
|---|---|---|---|---|
| KL | $-0.0077_{0.0937}$ | $0.1609_{0.0474}$ | $0.0111_{0.0133}$ | $0.0054_{0.0256}$ |
| KL + E | $0.1032_{0.0672}$ | $0.2756_{0.0216}$ | $0.0426_{0.0058}$ | $0.0680_{0.0123}$ |
| TVD | $0.0885_{0.0615}$ | $0.2699_{0.0666}$ | $0.0354_{0.0133}$ | $0.0605_{0.0202}$ |
| TVD+E | $0.1353_{0.0561}$ | $0.2712_{0.0602}$ | $0.0465_{0.0046}$ | $0.0837_{0.0081}$ |
| JSD | $0.0590_{0.0668}$ | $0.1763_{0.0445}$ | $0.0190_{0.0143}$ | $0.0307_{0.0231}$ |
| JSD+E | $0.0801_{0.0415}$ | $0.1955_{0.0513}$ | $0.0321_{0.0038}$ | $0.0542_{0.0066}$ |

Table 10: Difference ($\Delta$) in increase (+) or decrease (-) of metric score when calibrating with CROWD-CALIBRATOR compared to MaxProb $\hat{y}_{maj}$ for the **in-domain NLI test set**. We present the mean difference, with the standard deviation $\pm$ as subscript.

| | DAVIDSON | | FOUNTA | | HATECHECK | |
|---|---|---|---|---|---|---|
| | ΔAUC ↑ | ΔAUROC ↑ | ΔAUC ↑ | ΔAUROC ↑ | ΔAUC ↑ | ΔAUROC ↑ |
| Kamath et al. (2020) | $-0.0257_{0.0504}$ | $-0.0324_{0.0531}$ | $0.0001_{0.0188}$ | $-0.0086_{0.0154}$ | $-0.0407_{0.0798}$ | $0.0509_{0.0584}$ |
| Avg. Conf. - TVD+E** | $0.0217_{0.0085}$ | $0.0479_{0.0210}$ | $0.0202_{0.0084}$ | $0.0035_{0.0095}$ | $0.1554_{0.0146}$ | $0.2691_{0.0185}$ |
| Weighted Score - JSD+E** | $0.0129_{0.0051}$ | $0.0270_{0.0135}$ | $0.0210_{0.0077}$ | $0.0099_{0.0077}$ | $0.1168_{0.0079}$ | $0.1921_{0.0160}$ |
| DGHS - Label Dist. - TVD+E | $0.0164_{0.0083}$ | $0.0396_{0.0262}$ | $0.0061_{0.0056}$ | $0.0032_{0.0118}$ | $0.1471_{0.0156}$ | $0.2747_{0.0162}$ |

Table 11: Difference (Δ) in increase (+) or decrease (-) of metric score when calibrating with CROWD-CALIBRATOR compared to MaxProb $\hat{y}_{maj}$ for **unseen hate speech datasets**. We present the mean difference, with the standard deviation ± as subscript. For Davidson and HateCheck, we show the results when only using DGHS annotators and for Founta when only using GHC annotators (indicated with a **)

| | ANLI | | WANLI | | | | |
|---|---|---|---|---|---|---|---|
| | | | | cov@acc= ↑ | | | |
| | Δ AUC ↑ | Δ AUROC ↑ | Δ 0.7 | Δ 0.75 | Δ 0.8 | Δ AUC ↑ | Δ AUROC ↑ |
| KL | $0.0535_{0.0095}$ | $0.0782_{0.0216}$ | $-0.5097_{0.1309}$ | $-0.3578_{0.0343}$ | $-0.1034_{0.0872}$ | $-0.0567_{0.0073}$ | $-0.0906_{0.0136}$ |
| KL+E | $0.0217_{0.0084}$ | $0.0350_{0.0199}$ | $-0.0268_{0.0475}$ | $-0.0442_{0.0360}$ | $0.0352_{0.0685}$ | $0.0018_{0.0031}$ | $-0.0071_{0.0108}$ |
| TVD | $0.0456_{0.0107}$ | $0.0682_{0.0212}$ | $-0.2387_{0.0790}$ | $-0.2706_{0.0624}$ | $-0.0920_{0.0903}$ | $-0.0324_{0.0065}$ | $-0.0525_{0.0136}$ |
| TVD+E | $0.0202_{0.0071}$ | $0.0307_{0.0093}$ | $-0.0048_{0.0407}$ | $-0.0184_{0.0462}$ | $0.0325_{0.0825}$ | $0.0051_{0.0035}$ | $0.00026_{0.0098}$ |
| JSD | $0.0527_{0.0093}$ | $0.0744_{0.0201}$ | $-0.4908_{0.1290}$ | $-0.3594_{0.0347}$ | $-0.1026_{0.0868}$ | $-0.0498_{0.0068}$ | $-0.07342_{0.0139}$ |
| JSD+E | $0.0121_{0.0056}$ | $0.0171_{0.0057}$ | $0.0004_{0.0306}$ | $0.0182_{0.0399}$ | $0.0779_{0.0758}$ | $0.0092_{0.0028}$ | $0.0059_{0.0073}$ |

Table 12: Difference (Δ) in increase (+) or decrease (-) of metric score when calibrating with CROWD-CALIBRATOR compared to MaxProb $\hat{y}_{maj}$ for **unseen NLI datasets**. We present the mean difference, with the standard deviation ± as subscript.

# H   Analysis of Hate Speech Datasets

To measure the agreement and understand the dataset dynamic, we look at dataset characteristics of different hate speech datasets where we have access to who annotated what and each instance receives more than one annotation: HX (original in Figure 6 and binary version in Figure 7), MHSC (Figure 8), and GHC (Figure 9). The original HX dataset has three classes, *normal*, *offensive*, and *hate speech*. We then reduce it to a binary classification task. We plot the amount of comments annotated for each annotator and two metrics from CrowdTruth (Dumitrache et al., 2018): sentence quality and annotator/worker quality, which showcase the reliability and agreement in a dataset. Sentence quality describes the agreement of annotators for a given input and worker quality describes how much an annotator agrees with other annotators.

For all datasets, we see a similar story. The majority of the annotators have not rated many instances, except for GHC where many annotators have annotated more than 2000 comments. In general, we see many high-quality workers and a relatively smaller group of low-quality sentences. For HX, worker quality is lower when we keep offensive and hate speech separated.

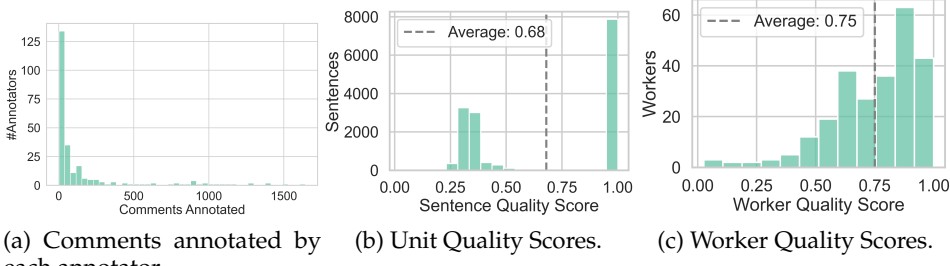

(a) Comments annotated by each annotator.

(b) Unit Quality Scores.

(c) Worker Quality Scores.

Figure 6: Dataset characteristics of HX dataset.

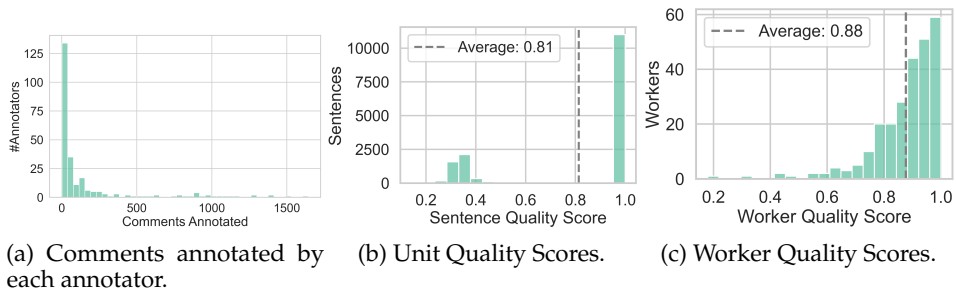

(a) Comments annotated by each annotator.

(b) Unit Quality Scores.

(c) Worker Quality Scores.

Figure 7: Dataset characteristics of HX dataset when reduced to binary classes.

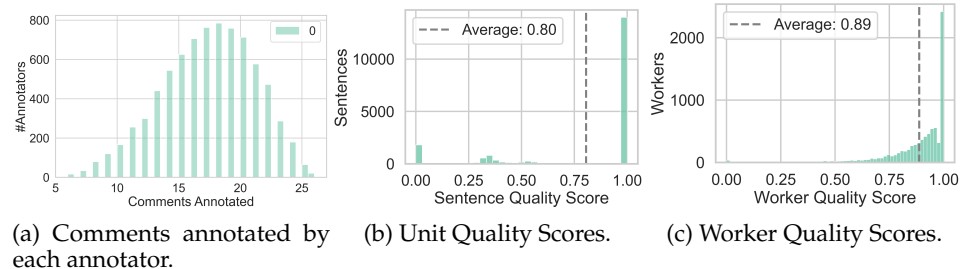

(a) Comments annotated by each annotator.

(b) Unit Quality Scores.

(c) Worker Quality Scores.

Figure 8: Dataset characteristics of MHSC dataset.

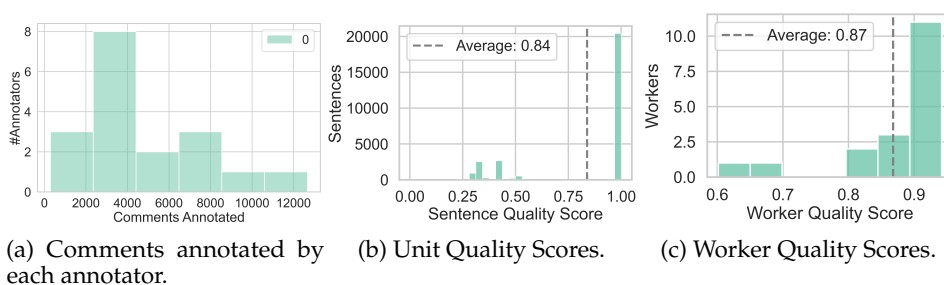

(a) Comments annotated by each annotator.

(b) Unit Quality Scores.

(c) Worker Quality Scores.

Figure 9: Dataset characteristics of GHC dataset.

