# OpenReview forum: "Crowd-Calibrator: Can Annotator Disagreement Inform Calibration in Subjective Tasks?"
_colmweb.org/COLM/2024/Conference — COLM_

### Official Review · Reviewer_jzn7 · 2024-05-10

**Rating:** 6
**Confidence:** 5
**Ethics Flag:** 1

**Summary:**

The authors propose a soft-labeling approach, along with the ability to abstain, to predictive modeling of subjective natural language tasks. They evaluate their model on two datasets HateXplain (HX)  Measuring Hate Speech Corpus (MHSC) combined into a single dataset against several baselines and another abstain model. They show that the baselines outperfom their model on in-domain test data, except when their abstention coverages is adjusted to be 90%, but that their model outperforms the others on out-of-domain data.

**Questions To Authors:**

Soft-labeling is a broader context than just modeling the responses of a population of raters. It can, for instance, also be used to model a rater's own level certainly. It might be good to clarify this. See also "Learning to Predict Population-Level Label Distributions" Liu et al. 2019 for another early example the soft-labeling approach.

**Reasons To Accept:**

+ The combination of soft-label with an abstention metric is novel. It was interesting to see how performance decreased as coverage increased!
+ out-of-domain performance is impresssive.
+ The paper is well-written.

**Reasons To Reject:**

- in-domain results are underwhelming, not so much because the authors's model only occasionally outperform the others but because of the inconsistency at different levels of coverage. Perhaps more test data is needed.

---

> ### Author Rebuttal · Authors · 2024-05-31
>
> We thank you for your review, constructive suggestions, and appreciation for finding the combination of soft label with abstention novel.
>
> **Inconsistency.** Our method especially works better when we aim to achieve higher accuracy, as the coverage is then higher than the baseline. The only case where this trend is broken is our Hate Speech dataset, where for Cov@Acc=0.9 our method gives the best coverage, but not for Cov@Acc=0.95. Additional experiments on the Online Misogyny task (as suggested by Var5, containing roughly similar amount of annotators) show slightly better performance than MaxProb with our Crowd Estimator for (baseline: 0.9205, ours: 0.9228) and AUROC (baseline: 0.7623, ours: 0.7741) and a somewhat similar pattern with more coverage for a higher accuracy. For Cov@Acc=0.85: baseline: 0.8875, ours: 0.8844 and Cov@Acc=0.9: baseline: 0.7094, ours: 0.7125. For Cov@Acc=0.95: baseline: 0.5125, ours: 0.5125.
>
> **Soft-labeling for raters.** Thank you for pointing this out, we will clarify this in the paper and cite Liu et al. 2019.

---

> > ### Comment · Reviewer_jzn7 · 2024-06-05
> > **Thanks for response.**
> >
> > nfm

---

### Official Review · Reviewer_efvr · 2024-05-11

**Rating:** 7
**Confidence:** 3
**Ethics Flag:** 1

**Summary:**

The paper introduces Crowd-Calibrator, a soft calibration method for subjective tasks (e.g. hate speech detection or NLI) that prevents a model to output prediction if its model confidence is far different than crowd label distribution. Experiment results show that the proposed method outperforms a selective prediction baseline (majority baseline + confidence thresholding) on unseen dataset.

**Questions To Authors:**

- Please provide definition/formula of the distance metrics used to better understand the results.
- How sensitive is the method on the nature of the task? Do you have any opinion on that? I guess one possible small experiments is to use two task in an almost identical setup (the same number of annotators) to isolate different variations of the training data you used.

**Reasons To Accept:**

- Well-executed experimental setup. I appreciate the authors in experimenting on different baselines, select the best one, and have evaluation on seen and unseen dataset.
- I also like that authors conduct experiments on different data situation, i.e. where there are few number vs. enough number of annotators. The setup is closer to real-world setup where we don't always have enough annotations/annotators which can give good insights for real-world experimentation.
- Paper is well-written and easy to follow.

**Reasons To Reject:**

While the authors provide thorough comparison on the various distance metrics and aggregation methods, it is hard to get the main takeaway from the overall results. Which distance metric (KL, TVD, JSD) should be used when applying this method? Does it depends on the nature of the datasets (# of annotators, task, etc). What are the effects of adding entropy (+E)?

---

> ### Author Rebuttal · Authors · 2024-05-31
>
> We thank you for the positive review and constructive suggestions!
>
> **Main takeaway:** Our results show that our method is beneficial for seen and unseen datasets when we have access to a small set of samples that have many annotations per instance (~100). Our results also show that our method can work on unseen datasets when we have access to individual annotator samples (> 2000). We will clarify these as the main takeaways of our paper.
>
> **Questions w.r.t. scoring strategies:** Our method offers flexibility in the choice of the distance metric, KL-divergence, JSD, and TVD (for which we will add the formulae in the Appendix) and aggregation strategy. In general, JSD and TVD give the best results in combination with the aggregation strategies of label distribution or averaged confidences. In all cases, it is beneficial to add entropy to the score of the metric as it prevents the model from still making decisions when both the model and crowds are uncertain of their response, while distance metrics alone would allow the model to make a prediction.
>
> **Sensitivity:** There does appear to be some sensitivity to the nature of the task, particularly the amount of annotations we can learn from for the Crowd Estimator. We see better performance when the Crowd Estimator learns from 100 annotations per sample in the case of NLI in comparison to estimating the crowd from 24 annotators. We’ll reflect on this in our discussion.

---

> > ### Comment · Reviewer_efvr · 2024-06-04
> >
> > Thanks for the clarification. I will keep my original score.

---

### Official Review · Reviewer_j6sj · 2024-05-14

**Rating:** 7
**Confidence:** 5
**Ethics Flag:** 1

**Summary:**

This paper proposes a new method called Crowd-Calibrator to calibrate models for subjective tasks leveraging disagreements among annotators. The method is based on the soft-label approach, to which it adds the idea of abstaining from the the selective prediction framework.  If a model's softmax output distribution is close to the human judgment distribution, the model makes a prediction; else the model abstains. The results suggest that the proposed method achieve better results than the baselines for the NLI domain, and for unseen texts in the hate speech domain.

**Questions To Authors:**

Questions:

1. Why didn't you use CE_soft as an evaluation metric?

2. Why didn't you compare your method with one of the standard soft-label methods (e.g., Peterson et al 2019/Uma et al 2020)?

References:

- there are many important papers making the point that disagreement is not necessarily going back at least to the work by Chklovsky and Mihalcea (2003) for wordsense disambiguation and Poesio et al (2006) for coreference, but the following papers at least should be cited if the paper is accepted:

Aroyo & Welty (2015). Truth Is a Lie: Crowd Truth and the Seven Myths of Human Annotation. AI Magazine, Vol. 36 No. 1

Rodrigues & Pereira (2018). Deep learning from crowds. In AAAI'18/IAAI'18/EAAI'18: Proceedings of the Thirty-Second AAAI Conference on Artificial Intelligence and Thirtieth Innovative Applications of Artificial Intelligence Conference and Eighth AAAI Symposium on Educational Advances in Artificial Intelligence, Pages 1611–1618

**Reasons To Accept:**

1. The research is aware of much of the relevant literature and was carried out in a methodological sound way.

2. The proposed method is interesting and, as far as I know, novel, and achieves good results on some datasets.

3. The comparative  analysis of model confidence when trained using various hard and soft loss functions in Section 2.3 is, as far as I know, novel and  very interesting.

**Reasons To Reject:**

1. Some of the evaluation questions are not properly justified. E.g., given that this is supposed to be a new soft-label method, but is it not compared with a standard soft-label model? Also, why isn't CE_soft used as a soft evaluation metric, given that it has become semi-standard in NLP (which is not to say it doesn't have its problems!)?

2. Some of the claims are a bit strong: e.g., in the abstract, 'we argue that that subjectivity should play a role in model decisions' is a bit strong given how many papers how come out in the last five-six years making the same point?

3. A number of methods which involve modelling the individual annotators have been proposed, most famously the Deep Learning from Crowd approach (which is not cited), and more recently the approach by Davani et al (which is), it would have been interesting to see why this particular approach was chosen.

---

> ### Author Rebuttal · Authors · 2024-05-31
>
> We thank you for your insightful comments and positive review! We answer your questions below.
>
> **Comparison to standard soft-label methods.** We compare our results to standard soft-label methods (Peterson et al. 2019/Uma et al. 2020) which use human disagreement to create soft labels $\hat{y}\_{soft}$ in Table 1. To make it compatible with abstention, we apply MaxProb. However, the results are far more conservative compared to using hard labels $\hat{y}\_{maj}$.
>
> **Related Work.** Thank you for the pointers; we will definitely cite these. Deep Learning from Crowd (DLC) is shown to be outperformed by learning soft labels with soft losses in Uma et al. 2020 and Uma et al. 2021, hence we did not include this method. Davani et al’s approach requires data where individual annotators have labeled many samples. This is something that many datasets and the ones we used in the paper---the HateXplain, Measuring Hate Speech Corpus, and NLI datasets—do not have. For the NLI datasets, we do not know who annotated what, we only have access to the distribution of labels per sample. Hence, we could not empirically compare to DwD; we will clarify this in our paper.
>
> **$CE_{soft}$ as an evaluation metric.** We evaluated the soft label model in Section 2.3 with Jensen-Shannon Divergence (JSD) as it is strictly more principled since unlike $CE_{soft}$ it is symmetrical. When evaluating for selective prediction, we do not use $CE_{soft}$ since our approach aims to improve calibration and selective prediction for which we use the widely used metrics.
>
> **Strong Claims.** Thank you for pointing this out, we agree and did not intend to suggest we are the first to point out that subjectivity should play a role, but we do want to restate its importance. We will make sure to add the references.

---

> > ### Comment · Reviewer_j6sj · 2024-06-04
> >
> > Good reply, still happy to accept the paper

---

### Official Review · Reviewer_Va5r · 2024-05-14

**Rating:** 6
**Confidence:** 4
**Ethics Flag:** 1

**Summary:**

The author proposes a novel method to calibrate models for subjective NLP classification tasks when crowdsourced annotations are available (rather than each instance only having a single ground truth). Here, "calibrate" refers to abstaining from making a prediction when the model is uncertain.

Specifically, a base model is trained on aggregated labels using the usual cross entropy loss (RoBERTa is used in this paper), then a classifier is trained for each worker. The worker classifier is formed by passing the embedding of the CLS token from the base model through a single hidden layer MLP. Subsequently, the output distributions from all worker models are aggregated into a crowd distribution. The distance between this crowd distribution and the base model's output distribution is calculated, and if the distance exceeds a threshold, the model chooses not to output a prediction. The author proposes several methods for how to aggregate worker distributions into a crowd distribution and how to calculate the distance. If the dataset only contains crowdsourced annotations but without worker information, then a crowd model is trained directly to fit each instance's label distribution, and the same method is used to calculate the distance between the crowd distribution and the base model's output distribution.

The author tested the proposed method and other baselines on the tasks of hate speech detection and natural language inference (NLI). The results showed competitive or superior performance.

**Questions To Authors:**

It’s not clear to me how the Weighted Scoring is calculated. Could the authors provide a formula for it?

**Reasons To Accept:**

Annotator disagreement in subjective NLP tasks is a prevalent issue that affects many applications.

The paper introduces a novel approach to calibrate models for subjective tasks, emphasizing the value of integrating human annotator insights into the model's decision-making process.

**Reasons To Reject:**

The author only conducted experiments on two tasks, where the NLI dataset has far more annotations per instance than typical ones, and the proposed model did not show superior performance to the baseline on the hate speech datasets. Additionally, in Table 2, the author chose different groups of workers to train worker models for each dataset, which also shows that the proposed model is quite sensitive to the choice of workers. In this case, the author should conduct more comprehensive testing of the proposed model on more datasets. For instance, a dataset for detecting online misogyny [1], and a dataset for multilabel emotion classification [2]. Moreover, comparisons should be made with some highly relevant works, such as [3], which also proposed training a model for each worker and discussed the calculation of uncertainty.

* [1] An Expert Annotated Dataset for the Detection of Online Misogyny (EACL 2021)
* [2] GoEmotions: A Dataset of Fine-Grained Emotions (ACL 2020)
* [3] Dealing with Disagreements: Looking Beyond the Majority Vote in Subjective Annotations (ACL 2022)

Although the paper primarily focuses on subjective NLP tasks, the chosen evaluation metrics are still based on the assumption that each instance has only one ground truth, such as accuracy in Cov@Acc=0.9. In fact, while reading the paper, I initially thought the evaluation would be about the ground truth being a distribution, the model also outputs a distribution, and then evaluating the performance by calculating the distance between the two distributions.

Another confusing point is "confidence." For a typical softmax output, like [0.4, 0.6], we would consider it represents p(y=0)=0.4, p(y=1)=0.6, or say the prediction is y=1 with confidence=0.6. However, in subjective tasks, if the ground truth is considered a distribution, then [0.4,0.6] just represents the predicted distribution by the model, which does not contain any information about confidence. It means the model might be very certain that the distribution is [0.4,0.6] or very uncertain. The unknown confidence can be calculated through other means, such as in the proposed method, if the base model's output distribution is very close to the crowd model's output, it is considered high confidence. However, in the paper, the term confidence seems to be interchangeably used with the model's output, as mentioned in the paragraph above Figure 4:
> "If the model confidence is far off from the crowd distribution, we want the model to abstain, i.e. not make a prediction."

But in Sec 4.1.2 it is mentioned:
> "If the model’s output distribution is far off from the human probability distribution, the model will abstain."

Clarifying these concepts more clearly could potentially reduce reader confusion.

---

> ### Author Rebuttal · Authors · 2024-05-31
>
> Thank you for your helpful comments and your appreciation of us tackling a crucial issue through a novel approach.
>
> **Additional Datasets:**  We chose our datasets specifically based on their complementary properties (also noted by j6sj) the most common real-world scenario with only limited annotators per sample (HS) and the ideal but rare scenario with many annotators per (NLI).
>
> Thanks for the dataset suggestions; these are similar to HS in having limited annotators per sample. We experimented with the suggested Online Misogyny dataset, and obtained results similar to HS: our Crowd-Calibrator slightly outperforms the MaxProb baseline in terms of AUC (baseline: 0.9205, ours: 0.9228) and AUROC (baseline: 0.7623, ours: 0.7741). We will add this in our paper.
>
> **Comparison with Davani et al. (DwD)** Davani et al.’s approach requires data where individual annotators have labeled many samples (as highlighted by their GoEmotions experiments). Most datasets including the HateXplain, Measuring Hate Speech Corpus, and NLI datasets do not follow this setting. For the NLI datasets, we only have access to the distribution of labels per sample. Hence, we could not empirically compare to DwD; we will clarify this in our paper.
>
> **Chosen evaluation metric**. We evaluate with widely used metrics for calibration (Brier’s Score) and selective prediction (Coverage@Accuracy, AUC, and AUROC) since our proposed setup is a soft approach for selective prediction and improved calibration. This does not mean we assume there is a single ground truth; rather we are observing a single sample from the underlying crowd distribution. Our crowd-calibrator abstains such that it does not predict samples that are far off from the crowd and where they both have high entropy.
>
> **Usage of “confidence”:** Thank you, we will update our paper with the term “softmax scores”.
>
> **Weighted Scoring formula.**
> The weighted scoring we used follows:
>
> $r_{non-hs} \cdot \text{score}(AvgConfidence_{non-hs}, \hat{y}\_{maj}) + r_{hs} \cdot \text{score}(AvgConfidence_{hs}, \hat{y}_{maj})$
>
> where $r_c$ is the ratio of votes for the particular class $c$. We will include this in the paper.

---

> > ### Comment · Reviewer_Va5r · 2024-06-04
> >
> > Thanks for the new results and the clarification. I've raised my score.

---

### Decision · Program_Chairs · 2024-07-10

**Decision:**

Accept

**Comment:**

This is a solid paper, on the topic of modelling subjective tasks and means of calibrating models to better reflect the aggregate human (crowd) distribution. The problem is novel, the work is well motivated, and the empirical evaluation convincing. Reviewers question how realistic it is to have datasets with rich crowd annotations, and thus the practical applicability of this work may be limited. However, this is not a reason to dismiss the paper: this line of work has the potential to influence future dataset annotation activities & release. The author responses helped to clarify a number of questions from reviewers (e.g., Va5r, who had several concerns, which were well addressed and the reviewer raised their score.)